# Luckiness in Multiscale Online Learning

**Muriel Felipe Pérez-Ortiz**
Centrum Wiskunde & Informatica (CWI)
muriel.perez@cwi.nl

**Wouter M. Koolen**
CWI and University of Twente
wmkoolen@cwi.nl

## Abstract

Algorithms for full-information online learning are classically tuned to minimize their worst-case regret. Modern algorithms additionally provide tighter guarantees outside the adversarial regime, most notably in the form of constant pseudoregret bounds under statistical margin assumptions. We investigate the multiscale extension of the problem where the loss ranges of the experts are vastly different. Here, the regret with respect to each expert needs to scale with its range, instead of the maximum overall range. We develop new multiscale algorithms, tuning schemes and analysis techniques to show that worst-case robustness and adaptation to easy data can be combined at a negligible cost. We further develop an extension with optimism and apply it to solve multiscale two-player zero-sum games. We demonstrate experimentally the superior performance of our scale-adaptive algorithm and discuss the subtle relationship of our results to Freund's 2016 open problem.

## 1 Introduction

The abstract problem of *online prediction with expert advice* [Littlestone and Warmuth, 1994, Freund and Schapire, 1997] is of fundamental importance in computational learning theory. Efficient and optimal algorithms for solving it have a substantial impact on various problems in general online convex optimization [Hazan, 2019], online model selection [Foster et al., 2017], boosting [Freund and Schapire, 1997], and maximal probabilistic inequalities [Rakhlin and Sridharan, 2017], to name a few. Concretely, a decision maker chooses among experts' advices sequentially, and the environment assigns each advice a scalar loss. If all losses have the same numerical range $[-\sigma, \sigma]$, the situation is well understood. Indeed, Freund and Schapire [1997] showed that, for $K$ experts and $t$ rounds, the Hedge algorithm guarantees the minimax regret (defined below) $\sigma\sqrt{2t \ln K}$. Furthermore, modern algorithms additionally guarantee lower or even constant regret when the sequence of losses is more benign [see De Rooij et al., 2014, Koolen and Van Erven, 2015, Mourtada and Gaïffas, 2019].

In the multiscale setting, where the experts' loss ranges may differ by orders of magnitude, it is natural to ask about the existence of algorithms that guarantee an optimal worst-case regret bound that scales with the loss range of the best expert instead of the maximum range. This question has been answered affirmatively [Chen et al., 2021, Bubeck et al., 2019, Cutkosky and Orabona, 2018, Foster et al., 2017]. The algorithms developed in this line of work have had a significant impact in different areas of computational learning theory and practice. Unfortunately, as we will see, the best known algorithms still fail to guarantee lower regret even for the simplest benign statistical cases. Ensuring these goals poses serious technical challenges. In particular, Bernstein's inequality, the engine of classical same-scale luckiness arguments, has no suitable multiscale upgrade. Moreover, intuitive candidate upgrades of same-scale results would contradict recent lower bounds (see Section 7). To make things worse, in order to obtain multiscale regret bounds, close attention needs to be paid to terms that are conventionally insignificant but now carry the maximum scale of the problem. This motivates our main question: *can a single algorithm have multiscale worst-case regret guarantees and, in addition, exhibit constant (pseudo)regret in stochastic lucky cases?*

36th Conference on Neural Information Processing Systems (NeurIPS 2022).

We answer the previous question affirmatively. The key contribution in this article is MUSCADA (multiscale adaptive), a computationally efficient algorithm that simultaneously guarantees a worst-case regret that grows with the scale of the best expert, and constant expected pseudoregret under a stochastic margin condition. MUSCADA uses a refined version of Follow the Regularized Leader based on the multiscale entropy of Bubeck et al. [2019]. Its crucial improvement is a second-order variance-like adaptation, the tightest possible for the analysis of this regularizer. This second-order adaptation is close in spirit to, and an improvement of, that of AdaHedge by De Rooij et al. [2014] and those of Chen et al. [2021]. As a result of careful analysis, MUSCADA has the following attractive properties: it does not need knowledge of the length of the game in advance without resorting to any doubling trick, the presence of zero-regret rounds does not change the state of the algorithm or its regret guarantees; it is invariant both under per-round, possibly unknown, translations of each expert's losses, and under a global known scaling common to all losses and ranges.

As an application of MUSCADA and its analysis techniques, we build an optimistic variant of the algorithm and use it to solve two-person zero-sum games that have a multiscale structure. The optimistic variant makes use of a guess of what the losses in the next round will be, and achieves lower regret when the guesses are adequate. This interest originates in the fact that optimistic algorithms converge to the solutions of such games at faster rates than their nonoptimistic counterparts [Syrgkanis et al., 2015]. We find experimentally that MUSCADA outperforms existing single-scale algorithms when the payoff matrix of the game exhibits a multiscale structure.

In the rest of this introduction we lay out formally the multiscale experts problem, review existing work, present a summary of the main contributions (Section 1.1), and outline the rest of the article.

**Full-information online learning.** In its simplest form, we must decide sequentially in rounds how to aggregate the predictions made by a fixed number $K$ of *experts*. At each round $t$, we choose an aggregation strategy, a probability distribution $\boldsymbol{w}_t \in \mathcal{P}(K)$ over experts. After choosing $\boldsymbol{w}_t$, we assess the quality of the experts' predictions with a numerical loss $\boldsymbol{\ell}_t = (\ell_{t,k})_{k \in K}$ and judge the performance of our aggregation strategy by the $\boldsymbol{w}_t$-weighted losses $\langle \boldsymbol{w}_t, \boldsymbol{\ell}_t \rangle = \sum_{k \in K} w_{t,k} \ell_{t,k}$. Our objective is to minimize the cumulative gap between the losses incurred by our aggregation strategy $t \mapsto \boldsymbol{w}_t$ and the best expert in hindsight. This cumulative gap is the *regret* $\mathcal{R}_t = \sum_{s=1}^t \langle \boldsymbol{w}_s, \boldsymbol{\ell}_s \rangle - \min_{k \in K} \sum_{s=1}^t \ell_{t,k}$. Other than range restrictions on the losses, no assumptions are made about the mechanism that generates them. More precisely, for each expert $k \in K$ and all rounds $t$, we only assume that $\ell_{k,t} \in [-\sigma_k, \sigma_k]$ for known nonnegative scales $\{\sigma_k\}_{k \in K}$. We call $\boldsymbol{R}_t$ the vector of regrets with respect to each expert, that is, the vector with entries $R_{t,k} = \sum_{s=1}^t \{\langle \boldsymbol{w}_s, \boldsymbol{\ell}_s \rangle - \ell_{s,k}\}$.

**Existing results.** Several algorithms have been proposed that achieve the worst-case regret in the multiscale setting, but none of them achieve constant regret in stochastic lucky cases. Motivated by the problem of online model selection, Foster et al. [2017] used a technique of adaptive relaxations to produce randomized algorithms that guarantee

$$\mathbf{E}_{\mathbf{P}}[R_{t,k}] = O\left(\sigma_k \sqrt{t(\ln t + \ln(1/\pi_k) + \ln(\sigma_k/\sigma_{\min}))}\right) \text{ as } t \to \infty,$$

where $\boldsymbol{\pi}$ is a prior distribution on experts that generalizes the uniform $1/K$ of the Hedge algorithm and the expectation is over the algorithm's randomness. Bubeck et al. [2019] first proposed a Follow-the-Regularized-Leader algorithm with a multiscale entropy regularization that guarantees

$$R_{t,k} = O\left(\sigma_k \sqrt{t(\ln K + \ln(\sigma_{\max}/\sigma_{\min}))}\right) \text{ as } t \to \infty$$

when the number of rounds $t$ is known in advance. Bubeck et al. [2019, Theorem 20] also construct an instance of the $K = 2$ experts problem in which there exists a time $t$ for which any algorithm must have $R_{t,k'} \gtrsim \sigma_{k'} \sqrt{t(\ln K + \ln(\sigma_{\max}/\sigma_{\min}))}$ for some expert $k'$, shedding some light on the minimax picture. Recently, Chen et al. [2021] designed an optimistic algorithm that uses the same regularization as Bubeck et al. [2019] with an additional ingredient: at each round, a second-order correction is added to the losses before computing the next round's weights. At every round, their algorithm makes use of a guess vector $\boldsymbol{m}_t$ that can depend on the losses up to time $t - 1$. The scale of the guesses $\boldsymbol{m}_t$ are assumed to be the same as that of the losses; $|m_{t,k}| \le \sigma_k$. For instance, valid choices for the guess $\boldsymbol{m}_t$ are $\boldsymbol{0}$ and the loss $\boldsymbol{\ell}_{t-1}$ of the previous round. The algorithm of Chen et al. [2021] achieves

$$R_{t,k} = O\left(\sigma_k \sqrt{\beta_{t,k} \ln t} + \sigma_{\max} \ln t\right) \text{ as } t \to \infty,$$

now scaling with the expert-dependent "time" $\beta_{t,k} = \sum_{s=1}^{t} \frac{(\ell_{s,k} - m_{s,k})^2}{\sigma_k^2} \leq 4t$. Furthermore, they show that a different single-scale tuning of their algorithm exhibits stochastic luckiness. Namely, if the losses are sampled from a distribution with a gap $d_{\min} > 0$ between the expected loss of the best expert $k^*$ and that of any other expert, their algorithm guarantees that

$$R_{t,k^*} = O_{\mathbf{P}} \left( \frac{\ln t}{d_{\min}} \right) \quad \text{as } t \to \infty,$$

where $\mathbf{P}$ is the distribution of the losses. Their technique for stochastic luckiness uses the upcoming learner's loss as the guess $m_{t,k} = \langle \boldsymbol{w}_t, \boldsymbol{\ell}_t \rangle$. Unfortunately, this approach cannot be extended to the multiscale case, as these guesses may violate the experts' loss ranges.

## 1.1 Main results

In this section we present succinctly the regret guarantees for MUSCADA. Firstly, we present multiscale worst-case regret guarantees. Secondly, we present the stochastic luckiness results and Massart's margin condition. We then prove analogs of these results for an optimistic modification of MUSCADA in Section 4. We close this introduction with an outline of the rest of the article.

**Worst-case bounds.** We propose two tunings for MUSCADA; they cover the cases where there is or is not an expert with loss range equal to zero. Our results imply Theorem 1.1 below; it contains the regret guarantees for MUSCADA, expressed in terms of $v_t$, an implicitly defined variance-like second-order data-dependent quantity. The quantity $v_t$, defined by the algorithm, is the tightest allowed by our analysis and enables our luckiness result, Theorem 3.1. We interpret $v_t$ through the upper bounds of Theorem 1.2, also below, as an internal scale-free measure of time, as $v_t \leq 4t$.

**Theorem 1.1** (Regret Bounds). *Consider* MUSCADA, $t \mapsto v_t$ *defined in Figure 1, and any initial probability distribution $\boldsymbol{\pi}$.*

- *If $\sigma_{\min} = \min_{k \in K} \sigma_k > 0$, Tuning 1 guarantees, for any loss sequence,*

$$R_{t,k} \leq c\, \sigma_k \sqrt{v_t(\ln(1/\pi_k) + \ln(\sigma_k/\sigma_{\min}))} + O(1) \quad \text{as } t \to \infty, \tag{1}$$

  *where $c$ is a constant depending only on $\boldsymbol{\pi}$. The constant $c$ is well-behaved: if $\max_{k \in K} \pi_k = 1 - \varepsilon$, then $c \leq 4\sqrt{2}(1 + 1/(2\ln(1+\varepsilon)))$.*

- *Even if $\min_{k \in K} \sigma_k = 0$, Tuning 2 ensures, for any loss sequence,*

$$R_{t,k} \leq 2\sigma_k \sqrt{2\, v_t(\ln(1/\pi_k) + \ln(1+v_t))}(1 + o(1)) \quad \text{as } t \to \infty. \tag{2}$$

The following theorem (proven in Appendix G) shows that $v_t$ is bounded by a second-order quantity. If $w_{t,k}$ are the weights played by MUSCADA at round $t$ and $\eta_{t-1,k}$ are its learning rates, $v_t$ is bounded by the variance over experts of the losses w.r.t. a tilted probability distribution $\tilde{w}_{t,k} \propto w_{t,k}\eta_{t-1,k}$. The shape of this quantity may seem surprising, but it is not artificial; our analysis shows that it is the tightest and, consequently, the natural second-order quantity associated to this choice of regularization. In Appendix G, we further motivate, via a Taylor approximation, the shape of the resulting upper bound.

**Theorem 1.2.** *Let $\tilde{w}_{t,k}$ be the probability distribution $\tilde{w}_{t,k} \propto w_{t,k}\eta_{t-1,k}$ and let $\Delta v_t = v_t - v_{t-1}$. Then, with either tuning from Figure 2, $v_t$, from Figure 1, satisfies*

$$\Delta v_t \leq 4\frac{\operatorname{var}_{\tilde{\boldsymbol{w}}_t}(\boldsymbol{\ell}_t)}{\langle \tilde{\boldsymbol{w}}_t, \boldsymbol{\sigma}^2 \rangle} \leq 4, \qquad \text{where} \qquad \operatorname{var}_{\tilde{\boldsymbol{w}}_t}(\boldsymbol{\ell}_t) = \langle \tilde{\boldsymbol{w}}_t, (\boldsymbol{\ell}_t - \langle \tilde{\boldsymbol{w}}_t, \boldsymbol{\ell}_t \rangle)^2 \rangle.$$

**Stochastic luckiness.** We now turn to our results for stochastic easy data. Not all stochastic scenarios are easy (in fact, worst-case regret lower bounds are proved using stochastic data). We use Massart's margin condition, a standard benchmark for easy data.

**Definition 1.3** (Massart's easiness condition). The losses $\boldsymbol{\ell}_1, \boldsymbol{\ell}_2, \ldots$ satisfy Massart's easiness condition if they are generated i.i.d. from a distribution $\mathbf{P}$ with the following property: there exists a constant $c_M$ and an expert $k^* \in K$ such that

$$\mathbf{E}_{\mathbf{P}}[(\ell_{t,k} - \ell_{t,k^*})^2] \leq c_M \mathbf{E}_{\mathbf{P}}[\ell_{t,k} - \ell_{t,k^*}]$$

for all $k \in K$ and $t \geq 1$. In that case, $k^* = \arg\min_{k \in K} \mathbf{E}_{\mathbf{P}}[\ell_{t,k}]$ for all $t$.

Massart's condition is implied by a more interpretable gap condition [Koolen et al., 2016, Lemma 3]. If there exist a gap $d_{\min} > 0$ in expectation between the loss of any expert and that of the best one $k^*$, that is, if, for every $k \neq k^*$, $\mathbf{E_P}[\ell_{1,k}] \geq d_{\min} + \mathbf{E_P}[\ell_{1,k^*}]$, Massart's condition is satisfied with $c_M = 1/d_{\min}$. We show the following theorem.

**Theorem 1.4** (Constant regret under Massart's condition). *Under Massart's condition (Definition 1.3),* MUSCADA *with either Tuning 1 or 2 has constant expected pseudoregret over time, that is,*

$$\mathbf{E_P}[R_{t,k^*}] \lesssim 1.$$

**Outline.** The rest of this article is organized as follows. In Section 2, we introduce and analyze MUSCADA. In Section 3, we state the main results on stochastic luckiness for MUSCADA. In Section 4, we introduce an optimistic variant of MUSCADA, give remarks about its numerical implementation in Section 5, and apply it to accelerating the solution of multiscale games in Section 6. We end this article with a discussion of our results in Section 7.

## 2 The MUSCADA Multiscale Online Learning Algorithm

In this section, we describe our algorithm and motivate its design. We present two useful tunings and prove the corresponding worst-case regret guarantees. For the sake of intuition, we specialize the algorithm to the case of same-scale experts with uniform prior and compare its resulting closed form to AdaHedge [De Rooij et al., 2014]. Stochastic luckiness results are found in Section 3. We begin by introducing some notation.

**Notation.** We use boldface type for vectors in $\mathbb{R}^K$ ($\boldsymbol{R}_t, \boldsymbol{L}_t, \boldsymbol{\mu}_t, \boldsymbol{\eta}_t, \boldsymbol{\sigma}, \boldsymbol{u}$) and distributions on $K$ experts ($\boldsymbol{p}, \boldsymbol{w}, \boldsymbol{\pi}$). We number rounds so that all quantities indexed by $t$ depend on the information witnessed by the learner in the first $t$ rounds. Exceptionally, we use weights $\boldsymbol{w}_t$ at round $t$. For two functions $f$ and $g$ we write "$f = O(g)$ as $t \to \infty$" if there exists $c > 0$ such that $\lim_{t \to \infty} f(t)/g(t) \leq c$. Similarly, we write "$f(t) \sim g(t)$ as $t \to \infty$" if $\lim_{t \to \infty} f(t)/g(t) = 1$, and $f \lesssim g$ if there is $c > 0$ so that $f \leq cg$. We denote the simplex of probability distributions on $K$ experts by $\mathcal{P}(K)$ and use $K$ interchangeably for a number $K \in \mathbb{N}$ and the set $\{1, \ldots, K\}$.

We define MUSCADA in Figure 1 and give its two main tunings in Figure 2. At round $t$, after observing cumulative corrected losses $\boldsymbol{L}_{t-1} + \boldsymbol{\mu}_{t-1}$, MUSCADA plays weights

$$w_{t,k} = u_k \mathrm{e}^{-\eta_{t-1,k}(L_{t-1,k} + \mu_{t-1,k} + a^*_{t-1})},$$

where $u_k > 0$ is a tuning parameter related to the prior weights, $\boldsymbol{\eta}_{t-1}$ are learning rates that decrease over time, $\boldsymbol{\mu}_t$ are corrections incrementally computed at every round, and the scalar $a^*_{t-1}$ ensures normalization (see Lemma F.7). The weights $\boldsymbol{w}_t$ are reminiscent of those played by the Hedge algorithm, but the normalization $a^*_t$ cannot be computed explicitly in general. The weights $\boldsymbol{w}_t$ are the result of a Follow-the-Regularized-Leader update on a vector of corrected losses $\boldsymbol{L}_{t-1} + \boldsymbol{\mu}_{t-1}$. The regularizer employed is the multiscale entropy: for a fixed $\boldsymbol{u} > 0$, its Bregman divergence is

$$\boldsymbol{w} \mapsto D_{\boldsymbol{\eta}}(\boldsymbol{w}, \boldsymbol{u}) = \sum_{k \in K} w_k \frac{\ln(w_k/u_k) - (1 - u_k/w_k)}{\eta_k}, \quad \boldsymbol{w} \in \mathcal{P}(K) \tag{3}$$

[see Bubeck et al., 2019, Chen et al., 2021]. The goal substracting the data-dependent second-order corrections $\boldsymbol{\mu}_t$ from the experts' regrets is to keep a scalar potential function $\Phi_t$ negative. Here, the potential $t \mapsto \Phi_t$ is defined by convex conjugacy with respect to the multiscale entropy as

$$\Phi_t := \Phi(\boldsymbol{R}_t - \boldsymbol{\mu}_t, \boldsymbol{\eta}_t) = \max_{\boldsymbol{w} \in \mathcal{P}(K)} \langle \boldsymbol{w}, \boldsymbol{R}_t - \boldsymbol{\mu}_t \rangle - D_{\boldsymbol{\eta}_t}(\boldsymbol{w}, \boldsymbol{u}), \tag{4}$$

for which $\boldsymbol{w}_{t+1}$ is the maximizer. The corrections $\boldsymbol{\mu}_t$ and the consequent negativity of the potential $\Phi_t$ are the main ingredients in the regret analysis of MUSCADA. We next motivate these choices.

**The shape of the corrections $\boldsymbol{\mu}_t$.** We designed MUSCADA to favor experts with low corrected regret $\boldsymbol{R}_t - \boldsymbol{\mu}_t$. For the sake of informal discussion, our goal is to obtain $\mu_{t,k} \approx \sigma_k \sqrt{v_t \ln(1/\pi_k)}$. The algorithm achieves this by additively correcting the regrets in each round. Indeed, from the analysis of entropy-regularized algorithms, one would expect learning rates of the shape $\eta_{t,k} \approx \frac{1}{\sigma_k} \sqrt{\frac{\ln(1/\pi_k)}{v_t}}$

---

**Parameters:** A vector $u_k > 0$ of initial weights, initial strictly positive learning rates $\eta_{0,k} \leq 1/(2\sigma_k)$, and real, continuous nonincreasing functions $H_k : \mathbb{R}^+ \mapsto \mathbb{R}$ with $H_k(0) = 1$.

**Initialization:** Let $\mu_{0,k} = 0$, $v_0 = 0$, $R_{0,k} = 0$ and $L_{0,k} = 0$. For each round $t = 1, 2, 3, \ldots$

1. Play (follow the multiscale-entropy regularized leader of the corrected losses)

$$\boldsymbol{w}_t = \operatorname*{arg\,min}_{\boldsymbol{w} \in \mathcal{P}(K)} \ \langle \boldsymbol{w}, \boldsymbol{L}_{t-1} + \boldsymbol{\mu}_{t-1} \rangle + D_{\boldsymbol{\eta}_{t-1}}(\boldsymbol{w}, \boldsymbol{u}), \tag{5}$$

   where $D_{\boldsymbol{\eta}}$ is the multiscale relative entropy given in (3).

2. Observe loss $\boldsymbol{\ell}_t$. Update $R_{t,k} = R_{t-1,k} + \langle \boldsymbol{w}_t, \boldsymbol{\ell}_t \rangle - \ell_{t,k}$ and $L_{t,k} = L_{t-1,k} + \ell_{t,k}$.

3. Compute $\Delta v_t$, the value $\Delta v \geq 0$ such that

$$\Phi(\boldsymbol{R}_t - \boldsymbol{\mu}_{t-1} - \boldsymbol{\sigma}^2 \boldsymbol{\eta}_{t-1} \Delta v, \boldsymbol{\eta}_{t-1}) = \Phi(\boldsymbol{R}_{t-1} - \boldsymbol{\mu}_{t-1}, \boldsymbol{\eta}_{t-1}), \tag{6}$$

   where $\Phi$ is the potential function defined in (4).

4. Compute $\Delta \mu_{t,k} = \sigma_k^2 \eta_{t-1,k} \Delta v_t$. Update $\mu_{t,k} = \mu_{t-1,k} + \Delta \mu_{t,k}$ and $v_t = v_{t-1} + \Delta v_t$.

5. Set the new learning rate $\eta_{t,k} = \eta_{0,k} H_k(v_t)$.

---

Figure 1: MUSCADA

to be optimal. With this learning rates in mind, the desired correction $\boldsymbol{\mu}_t$ can be approximated using a Riemann-sum approximation of $\sqrt{v_t} = \int_0^{v_t} \frac{1}{2\sqrt{v}} \mathrm{d}v$. Indeed, for the conjectured learning rates, our target $\mu_{t,k}$ satisfies $\mu_{t,k} \approx \sigma_k^2 \sum_{s \leq t} \eta_{s-1,k} \Delta v_s$, where $\Delta v_t = v_t - v_{t-1}$. This implies that the choice $\Delta \mu_{t,k} = \sigma_k^2 \eta_{t-1,k} \Delta v_t$ as our per-round additive correction is helpful for achieving our goal. We discuss our precise choice of learning rates after the formal statement of Proposition 2.2 below.

**Negativity of $\Phi$.** Our regret bounds are a direct consequence of the negativity of the potential $t \mapsto \Phi_t$. Indeed, by its definition, $\Phi_0 \leq 0$, and, because of our choice of nonincreing learning rates and corrections, the change in potential $\Delta \Phi_t = \Phi_t - \Phi_{t-1}$ can be bounded by

$$\Delta \Phi_t \leq \Phi(\boldsymbol{R}_t - \boldsymbol{\mu}_t, \boldsymbol{\eta}_{t-1}) - \Phi(\boldsymbol{R}_{t-1} - \boldsymbol{\mu}_{t-1}, \boldsymbol{\eta}_{t-1}) = 0,$$

where the last equality follows from (6), the choice of corrections $\Delta \boldsymbol{\mu}_t$. This implies the following lemma, of which we give a more general proof in Section C.1.

**Lemma 2.1.** The potential $t \mapsto \Phi_t$ starts at $\Phi_0 \leq 0$ and is decreasing for $t \geq 0$.

Once we prove that the potential $\Phi_t$ is negative, we are ready to derive regret guarantees for MUS-CADA. The maximal nature of the potential $t \mapsto \Phi_t$ and its nonpositivity together imply that, *simultaneously* for all distributions $\boldsymbol{p} \in \mathcal{P}(K)$,

$$\langle \boldsymbol{p}, \boldsymbol{R}_t - \boldsymbol{\mu}_t \rangle \leq D_{\boldsymbol{\eta}_t}(\boldsymbol{p}, \boldsymbol{u}). \tag{7}$$

We choose $\boldsymbol{p}$ concentrated on each expert $k \in K$ to deduce the next proposition (proof in Section C.1).

**Proposition 2.2.** *Assume that the learning rates $t \mapsto \boldsymbol{\eta}_t$ are decreasing.* MUSCADA *guarantees that, for any $t = 1, 2, 3, \ldots$ and all $k \in K$,*

$$R_{t,k} \leq \mu_{t,k} + \frac{\ln(1/u_k)}{\eta_{t,k}} + \sum_{j \in K} \frac{u_j}{\eta_{t,j}} - \frac{1}{\eta_{t,k}}, \tag{8}$$

*where $\mu_{t,k} = \sigma_k^2 \sum_{s \leq t} \eta_{s-1,k} \Delta v_s$. Furthermore, for $\eta_{t,k} = \eta_0 H_k(v_t)$ as in Figure 1, $\boldsymbol{\mu}_t$ satisfies*

$$\mu_{t,k} \leq \sigma_k^2 \eta_{0,k} \int_0^{v_t} H_k(x) \mathrm{d}x + \sigma_k^2 (\eta_{0,k} - \eta_{t,k}) \max_{s \leq t} \Delta v_s. \tag{9}$$

**Choice of learning rates.** Proposition 2.2 guides us in choosing the learning rates presented in Figure 2. The starting value of the learning rates influences our ability to control $v_t$ in terms of the variance of the losses of the algorithm while their behavior for large $v_t$ determines the long-term growth of the regret bounds. The learning rates presented in Figure 2 interpolate smoothly

Let $\boldsymbol{\pi} \in \mathcal{P}(K)$ be a probability distribution on $K$ experts.

**Tuning 1** Requires $\sigma_{\min} > 0$. Set $u_k = \pi_k \frac{\sigma_{\min}}{\sigma_k}$, $\eta_{0,k} = \frac{1}{2\sigma_{\max}}$, $\gamma_k = 8\frac{\sigma_{\max}^2}{\sigma_k^2}\ln(1/u_k)$ and

$$H_{1,k}(v) = \frac{\mathrm{d}}{\mathrm{d}v}\left[\frac{v}{\sqrt{1+v/\gamma_k}}\right] = \frac{v/\gamma_k + 2}{2(1+v/\gamma_k)^{3/2}}.$$

**Tuning 2** Set $u_k = \pi_k$, $\eta_{0,k} = \frac{1}{2\sigma_{\max}}$, $\alpha_k = 32\frac{\sigma_{\max}^2}{\sigma_k^2}$, $\gamma_k = \alpha_k \ln(1/u_k)$ and

$$H_{2,k}(v) = \frac{\mathrm{d}}{\mathrm{d}v}\left[\sqrt{\alpha_k^2\left\{(1+v/\alpha_k)\ln(1+v/\alpha_k) - v/\alpha_k\right\} + \frac{v^2}{2(1+v/(2\gamma_k))}}\right]$$

$$= \frac{\alpha_k \ln(1+v/\alpha_k) + \frac{1}{2}\frac{2v+v^2/(2\gamma_k)}{(1+v/(2\gamma_k))^2}}{2\sqrt{\alpha_k^2\left\{(1+v/\alpha_k)\ln(1+v/\alpha_k) - v/\alpha_k\right\} + \frac{v^2}{2(1+v/(2\gamma_k))}}}.$$

If, for some $k$, $\sigma_k = 0$, define $H_{2,k}$ to be the limit value $\lim_{\sigma\downarrow 0} H_{2,k}(v_t) = 1$.

Figure 2: Tunings

between these two regimes by taking the form $\eta_{t,k}^{(1)} = \eta_{0,k}H_{1,k}(v_t)$ and $\eta_{t,k}^{(2)} = \eta_{0,k}H_{2,k}(v_t)$. Here, the starting learning rates are set to $\eta_{0,k} = 1/(2\sigma_{\max})$. The functions $H_{1,k}, H_{2,k} \leq 1$ decrease monotonically from their initial values $H_{1,k}(0) = H_{2,k}(0) = 1$ in such a way that, as $v_t \to \infty$,

$$\eta_{t,k}^{(1)} \sim \frac{\sqrt{2}}{\sigma_k}\sqrt{\frac{\ln(1/\pi_k)}{v_t}} \qquad \text{and} \qquad \eta_{t,k}^{(2)} \sim \frac{\sqrt{2}}{\sigma_k}\sqrt{\frac{\ln(1/\pi_k) + \ln v_t}{v_t}}.$$

The asymptotic expresion for $\eta_{t,k}^{(1)}$ is reminiscent of the optimal learning rates for the Hedge algorithm with the number of rounds $t$ replaced by the refined $v_t$ and the uniform $\ln K$ replaced by $\ln(1/\pi_k)$. Finally, with the Riemann sum bound (9) from Proposition 2.2 in mind, the learning rates were chosen as the derivatives of functions that will become the dominant term in the regret guarantees.

**Tuned regret bounds.** The learning rates from Figure 2 can be readily used in Proposition 2.2 to derive regret guarantees for MUSCADA. However, to facilitate interpretation, we bound the learning rates and their reciprocals in order to obtain the regret bounds contained in the following proposition (proof in Appendix C.2). After its statement, we prove Theorem 1.1 from the introduction.

**Proposition 2.3.** *Let $\boldsymbol{\pi}$ be a probability distribution on $K$.*

- MUSCADA *run with Tuning 1 depicted in Figure 2 guarantees that, for any $t = 1, 2, \ldots$,*

$$R_{t,k} \leq 2\sigma_k\sqrt{2v_t \ln(1/u_k)} + c_{\boldsymbol{\sigma},\boldsymbol{\pi}}\sigma_{\min}\sqrt{2v_t} + 8\sigma_{\max}\ln(1/u_k) + 4\sigma_{\max} + \frac{\sigma_k}{2}\max_{s\leq t}\Delta v_s, \quad (10)$$

  *where the constant $c_{\boldsymbol{\sigma},\boldsymbol{\pi}} = \sum_{k\in K}\pi_k(1/\sqrt{\ln(1/u_k)})$ and $u_k = \pi_k\frac{\sigma_{\min}}{\sigma_k}$.*

- MUSCADA *run with Tuning 2 depicted in Figure 2 guarantees that, for any $t = 1, 2, \ldots$,*

$$R_{t,k} \leq 2\sigma_k\sqrt{2v_t\left(\ln\left(1+\frac{\sigma_k^2 v_t}{32\sigma_{\max}^2}\right) + \ln(1/\pi_k)\right)} + \sigma_k\ln(1/\pi_k)Z_k + \sum_{j\in K}\pi_j\sigma_j Z_j + \frac{\sigma_k}{2}\max_{s\leq t}\Delta v_t,$$

$$(11)$$

  *where $Z_k = \sqrt{\dfrac{v_t}{2\ln\left(1+\frac{\sigma_k^2 v_t}{32\sigma_{\max}^2}\right)}}\left(1 + \sqrt{\dfrac{\min\{\ln(1/\pi_k), \frac{\sigma_k^2 v_t}{16\sigma_{\max}^2}\}}{\ln\left(1+\frac{\sigma_k^2 v_t}{32\sigma_{\max}^2}\right)}}\right) = O\left(\sqrt{\dfrac{v_t}{\ln v_t}}\right)$ as $v_t \to \infty$.*

*Proof of Main Theorem 1.1.* With Proposition 2.3 at hand, we can prove the claims made in Section 1.1. Use the fact that $\sigma_{\min} \le \sigma_k$ to conclude from (10) that, as $t \to \infty$,

$$R_{t,k} \le 2\sigma_k \sqrt{2v_t \ln(1/u_k)} + 2c_{\boldsymbol{\sigma},\boldsymbol{\pi}} \sigma_k \sqrt{2v_t} + O(1).$$

We can bound $c_{\boldsymbol{\sigma},\boldsymbol{\pi}}/\sqrt{\ln(1/u_k)} \le 1/\ln(1/\pi_{\max})$, where $\pi_{\max} = \max_{k \in K} \pi_k$. Consequently,

$$R_{t,k} \le 2\sigma_k \left\{ 1 + 1/(2 \ln(1 + \varepsilon)) \right\} \sqrt{2v_t \ln(1/u_k)} + O(1)$$

as $t \to \infty$ any time that $\pi_{\max} = 1 - \varepsilon$. This coincides with (1). Similarly, (11) implies (2). $\qquad\square$

## 2.1 Closed-form solutions in the single-scale uniform-prior case

To help in the interpretation and to illustrate the challenges of the multiscale problem, we instantiate MUSCADA to a situation where all calculations can be carried out in closed form: when all scales are the same and equal to $\sigma$, and the initial weights $\boldsymbol{\pi}_{\mathrm{Unif}}$ are uniform; $\pi_{\mathrm{Unif},k} = 1/K$. This is the setting in which AdaHedge by De Rooij et al. [2014] operates. In this case, the learning rates and corrections of MUSCADA are the same for all experts; $\eta_{t,k} = \eta_t$ and $\Delta\mu_{t,k} = \Delta\mu_t$. The potential $\Phi_t$ and the corrections $\Delta\mu_t$ take the familiar form

$$\Phi_t = \frac{1}{\eta_t} \ln\left( \frac{1}{K} \sum_{k \in K} e^{\eta_t (R_{t,k} - \mu_{t,k})} \right), \qquad \text{and} \qquad \Delta\mu_t = \frac{1}{\eta_{t-1}} \ln \sum_{k \in K} w_{t,k} e^{\eta_{t-1}(\langle \boldsymbol{w}_t, \boldsymbol{\ell}_t \rangle - \boldsymbol{\ell}_t)}.$$

These two quantities play a central role in the analysis of AdaHedge, where De Rooij et al. [2014] called $\Delta\mu_t$ the *mixability gap*, the difference between the average $\langle \boldsymbol{w}_t, \boldsymbol{\ell}_t \rangle$ and the *mixed average* $-\frac{1}{\eta_{t-1}} \ln \sum_{k \in K} w_{t,k} e^{-\eta_{t-1}\ell_{t,k}}$. The main quantity in our analysis, $\Delta v_t$, becomes

$$\Delta v_t = \frac{1}{\eta_{t-1}^2 \sigma^2} \ln \sum_{k \in K} w_{t,k} e^{\eta_{t-1}(\langle \boldsymbol{w}_t, \boldsymbol{\ell}_t \rangle - \ell_{t,k})}.$$

Using well-known estimates for cumulant generating functions, $\Delta v_t$ can be bounded by the ratio $\mathrm{var}_{\boldsymbol{w}_t}(\boldsymbol{\ell}_t)/\sigma^2$. Indeed, Hoeffding's inequality implies the worst-case bound $\Delta v_t \le \frac{1}{2}$; Bernstein's, the second-order $\Delta v_t \lesssim \mathrm{var}_{\boldsymbol{w}_t}(\boldsymbol{\ell}_t)/\sigma^2$. Since it is $v_t$ that appears in the regret bounds in Proposition 2.3, they are a refinement over those of AdaHegde[1]. Additionally, the present analysis yields improvements that are apparent in lower-order terms. Indeed, the last two terms in the regret bound (8) in Proposition 2.2 vanish, and the analysis used in the proof of Proposition 2.3 with $\eta_0 = \sqrt{2}/\sigma$ and the instantiation of $H_1$ from Figure 2, $H_1(x) = \frac{x/\ln(K) + 2}{2(1 + x/\ln(K))^{3/2}}$, give the regret bound

$$\mathcal{R}_t \le \begin{cases} c_1 \sigma v_t + c_2 \sigma \ln K + \sigma/2 & \text{if } v_t \le \ln K, \\ 2\sigma\sqrt{2v_t \ln K} + \sigma/2 & \text{if } v_t > \ln K \end{cases}$$

with $c_1 = 3/\sqrt{2}$ and $c_2 = 1/\sqrt{2}$. Unfortunately, multiscale analogs of Bernstein and Hoeffding's inequalities on $\Delta v_t$ are not available; considerably more technical work needs to be carried out to prove Theorem 1.2. A multiscale analog of Bernstein's estimate for $\Delta v_t$ is only available when all the learning rates are smaller than $1/(2\sigma_{\max})$ (see the proof of Theorem 1.2 in Appendix G).

## 3 Multiscale Stochastic Luckiness

In this section we show, under easiness conditions, that the expected pseudoregret of MUSCADA is constant. Assume that the loss vectors $\boldsymbol{\ell}_1, \boldsymbol{\ell}_2, \ldots$ are i.i.d. and are generated according to a distribution $\mathbf{P}$ that satisfies Massart's easiness condition (see Definition 1.3). For Tuning 1, assume that the minimum scale among experts $\sigma_{\min}$ is strictly positive. The analysis technique in this case is similar to that of Koolen et al. [2016] with an extra step. A use of Theorem 1.2 shows that $\Delta v_t$ can be estimated in terms of $\mathrm{var}_{\boldsymbol{w}_t}(\boldsymbol{\ell}_t)$. This estimate possibly incurs in a multiplicative factor that can be as high as $1/\sigma_{\min}^2$. There are examples for which this constant is necessary (not shown). After this, standard arguments show that the expected pseudoregret is constant. See Appendix E for proofs.

---

[1] Our algorithm with learning rate tuning function $H(v) = \sqrt{\frac{\ln K}{4v}}$ comes closest to AdaHedge.

**Theorem 3.1.** *Under Massart's condition and using Tuning 1 from Figure 2, the expected pseudoregret of* MUSCADA *is bounded by a constant in the number of rounds. Specifically, for any* $t \geq 0$,

$$\mathbf{E}_{\mathbf{P}}[R_{t,k^*}] \lesssim a^2 c_{\mathrm{M}} + b,$$

*where* $a = \sqrt{2 \max_{i,j \in K} \left\{ \frac{1}{\sigma_i \sigma_j} \frac{\ln(1/\pi_i) + \ln(\sigma_i/\sigma_{\min})}{\ln(1/\pi_j) + \ln(\sigma_j/\sigma_{\min})} \right\}} \left( 4\sigma_{k^*} \sqrt{2\ln(1/u_{k^*})} + 2\sqrt{2} c_{\boldsymbol{\sigma},\boldsymbol{\pi}} \sigma_{\min} \right)$ *and* $b = 8\sigma_{\max} \ln(1/u_{k^*}) + 4\sigma_{\max} + 2\sigma_{k^*}$.

For Tuning 2, where we do not assume that $\sigma_{\min} > 0$, still $\mathbf{E}_{\mathbf{P}}[R_{t,k^*}] \lesssim 1$ using a different proof technique. Using the expression for the weights of the algorithm, we show that they concentrate on the best expert $k^*$. The analysis here is similar to that of Mourtada and Gaïffas [2019], but the lack of an expression for the normalizing $a_t^*$ presents with an additional technical difficulty. The result is the following theorem.

**Theorem 3.2.** *Let* $d_k = \mathbf{E}_{\mathbf{P}}[\ell_{t,k} - \ell_{t,k^*}]$ *and assume that* $\min_{k \neq k^*} d_k > 0$. *Using Tuning 2 in Figure 2,* MUSCADA *guarantees constant expected pseudoregret. Specifically,*

$$\mathbf{E}_{\mathbf{P}}[R_{t,k^*}] \leq \sum_{k \in K} f(d_k), \qquad \text{where} \qquad f(d) = O\left( \frac{\sigma_{\max}^2}{d} \ln\left( \frac{\sigma_{\max}^2}{d^2} \right) \right) \text{ as } d \to 0.$$

Standard modifications of the arguments presented may be used to prove that the pseudoregret is constant with $\mathbf{P}$-high probability (not shown).

## 4  Optimism

In this section we show an optimistic variant of MUSCADA. Suppose that, before round $t$, we count on guesses $\boldsymbol{m}_t$ for what $\boldsymbol{\ell}_t$ will be. Assume that $\boldsymbol{m}_t$ is of the same scale as $\boldsymbol{\ell}_t$, that is, $|m_{t,k}| \leq \sigma_k$. In particular, this entails that $|\ell_{t,k} - m_{t,k}| \leq 2\sigma_k$. A modification of MUSCADA, presented in Figure 1, puts these guesses to good use. These modifications allow for regret guarantees similar to those contained in Proposition 2.3, but in this case $\Delta v_t^\circ \lesssim \mathrm{var}_{\tilde{\boldsymbol{w}}_t^\circ}(\boldsymbol{\ell}_t - \boldsymbol{m}_t)/\langle \tilde{\boldsymbol{w}}_t^\circ, \boldsymbol{\sigma}^2 \rangle$, where the superscript $\circ$ signals the optimistic analogs of the quantities from MUSCADA. These modifications are shown in Figure 3 and the regret bounds in the following proposition (proofs in Appendix D).

**Proposition 4.1.** *If* $t \mapsto v_t^\circ$ *is the variance process defined by Optimistic* MUSCADA *in Figure 3, the same regret bounds presented Proposition 2.3 hold with two modifications:* $v_t^\circ$ *instead of* $v_t$ *and all scales doubled, that is,* $2\boldsymbol{\sigma}$ *instead of* $\boldsymbol{\sigma}$. *Furthermore, for each* $t = 1, 2, \dots$, $\Delta v_t^\circ \leq 4 \mathrm{var}_{\tilde{\boldsymbol{w}}_t^\circ}(\boldsymbol{\ell}_t - \boldsymbol{m}_t)/\langle \tilde{\boldsymbol{w}}_t^\circ, \boldsymbol{\sigma}^2 \rangle \leq 4t$, *where* $\tilde{w}_{t,k}^\circ \propto w_{t,k}^\circ \eta_{t-1,k}$.

---

1'  Compute the guess $\boldsymbol{m}_t$ and play

$$\boldsymbol{w}_t^\circ = \underset{\boldsymbol{w} \in \mathcal{P}(K)}{\arg\min} \langle \boldsymbol{w}, \boldsymbol{L}_{t-1} + \boldsymbol{m}_t + \boldsymbol{\mu}_{t-1} \rangle - D_{\boldsymbol{\eta}_{t-1}}(\boldsymbol{w}, \boldsymbol{u}).$$

3'  Let $\Delta v_t^\circ$ be the value $\Delta v^\circ \geq 0$ such that

$$\Phi(\boldsymbol{R}_t - \boldsymbol{\mu}_{t-1} - \boldsymbol{\eta}_{t-1}\boldsymbol{\sigma}^2 \Delta v^\circ, \boldsymbol{\eta}_{t-1}) = \Phi(\boldsymbol{R}_{t-1} + \langle \boldsymbol{w}_t^\circ, \boldsymbol{m}_t \rangle - \boldsymbol{m}_t - \boldsymbol{\mu}_{t-1}, \boldsymbol{\eta}_{t-1}). \quad (12)$$

**Tuning 1' and Tuning 2'.**  As in Figure 2 but with halved starting learning rate $\eta_{0,k} = \frac{1}{4\sigma_{\max}}$.

---

Figure 3: Optimistic MUSCADA, given as update w.r.t. Figure 1.

## 5  Computation

At each round, MUSCADA requires two computations. We now argue that both can be executed to machine precision in $O(K)$ time. First, computing the weights (5) given the losses $\boldsymbol{L}_{t-1}$ and correction terms $\boldsymbol{\mu}_{t-1}$ can be reduced, by Lemma F.6, to a single scalar convex minimization problem. Cancelling the derivative of the objective amounts to searching for the normalizing offset $a_t$. To that

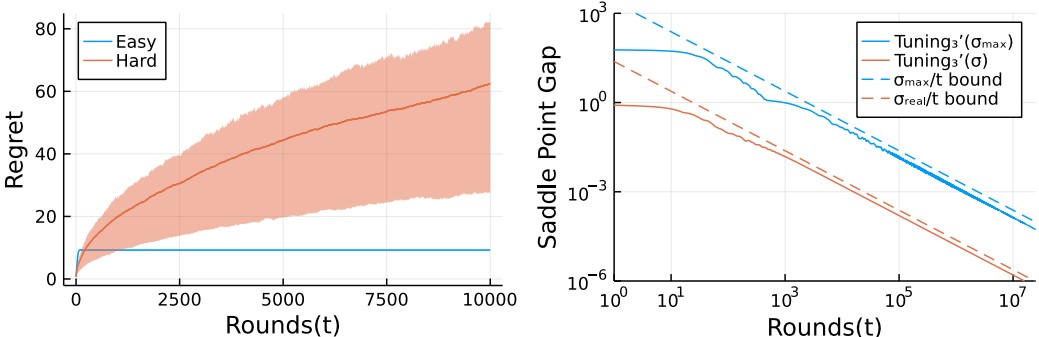

Figure 4: Left: empirical mean and quartiles of 2000 realizations of the regret $t \mapsto R_{t,k^*}$ of MUSCADA. For easy i.i.d. Massart distribution, the regret is constant; for a hard distribution without a gap, $\Omega(\sqrt{t})$. Right: optimistic MUSCADA (solid red) achieves an iterate-average saddle-point gap of $\sigma_{\mathrm{real}}/t$ where $\sigma_{\mathrm{real}} = \sigma_{\mathrm{max}}/100$ is the relevant scale of the Nash equilibrium. Other methods scale as $\sigma_{\mathrm{max}}/t$.

end, binary search to machine precision takes $O(K)$ time per round. Notice that this also allows us to compute the potential value. Second, for computing the variance contribution (6), we observe that the right hand side of (6) is decreasing in $\Delta v_t$. Since the potential can be computed in $O(K)$ time, we can use an outer binary search to compute $\Delta v_t$ to machine precision in $O(K)$ time as well. Alternatively, Newton's method may be employed; both of the previous problems require finding a root of a convex function. When deferring to a convex optimization library, a convenient expression is the jointly convex minimization form (see Lemma F.6)

$$\Delta v_t = \inf_{a, \Delta v} \Delta v \quad \text{subject to} \quad a + \sum_{k \in K} w_{t,k} \frac{e^{\eta_{t-1,k}(\langle \boldsymbol{w}_t, \boldsymbol{\ell}_t \rangle - \ell_{t,k} - a) - \eta_{t-1,k}^2 \sigma_k^2 \Delta v} - 1}{\eta_{t-1,k}} \leq 0.$$

## 6  Experiments on Synthetic Data

We investigate the performance of our multiscale method on two experiments: one for illustrating the performance of MUSCADA under Massart's condition, another for solving multiscale two-player zero-sum games.

The aim of the first experiment is to compare the performance of MUSCADA in easy and hard stochastic data sequences. To this end, we compared a sequence of hard stochastic data with no gap vs. easy data sampled i.i.d. from a distribution satisfying Massart's condition. We witnessed constant regret for the easy data, as shown in Figure 4 (Left). We take $K = 50$ experts and set $\sigma_k = 1/k$ for each $k \in K$. To generate our data, we fix some mean $\lambda_k \in [-\sigma_k, \sigma_k]$ and generate binary expert losses $\ell_{t,k} \in \{-\sigma_k, +\sigma_k\}$ independently between rounds and experts, with probability $\mathbf{P}\{\ell_{t,k} = \sigma_k\} = \frac{\sigma_k + \lambda_k}{2\sigma_k}$. For the hard case, we set $\lambda_k = 0$ for all $k$. For the lucky case, we set $\lambda_2 = -1/5$ instead. Generating this figure with the code in the supplementary material takes 3 seconds on an Intel i7-7700 processor.

The aim of the second experiment is to show the performance of MUSCADA for solving multiscale zero-sum games. Here, the payoff matrix is unknown, but row and column scales are available and vastly different. As detailed in Appendix A, we run two instances of appropriately tuned Optimistic MUSCADA against each other. As shown in Figure 4 (Right), the pair of time-average iterates converges to the saddle point with a suboptimality gap of order $\sigma_{\mathrm{real}}/t$ instead of the worst-case $\sigma_{\mathrm{max}}/t$, where $\sigma_{\mathrm{real}}$ is the maximum range within the support of the saddle point. In Appendix A, we conjecture that this rate holds for any such game and prove a weaker result: without optimism, the slower but scale-adaptive rate $\sigma_{\mathrm{real}}/\sqrt{t}$ is achieved.

# 7 Discussion

We developed a new algorithm for multiscale online learning that is both worst-case safe and achieves constant pseudoregret in stochastic lucky cases. Our method is a refinement of the Follow-the-Regularized-Leader template with a weighted entropy. The main innovation is in the correction terms added to the losses, which are the tightest the technique admits. This suggests that these variance-like terms are in fact intrinsic to the problem of obtaining scale-dependent regret bounds. Lastly, we relate this newfound variance to the variance asked for by Freund [2016], we comment on the advantage of second-order guarantees over zeroth-order ones, and we state an open problem.

**Quantile bounds and solving Freund's problem.** Freund [2016] asked whether quantile adaptivity and variance adaptivity are compatible, that is, whether one can have $\langle \boldsymbol{p}, \boldsymbol{R}_t \rangle \leq \sqrt{\mathrm{KL}(\boldsymbol{p}, \boldsymbol{u}) \sum_{s \leq t} \mathrm{var}_{\boldsymbol{w}_s}(\boldsymbol{\ell}_s)}$ for all comparator distributions $\boldsymbol{p} \in \mathcal{P}(K)$ simultaneously. Even though our tuning of $\boldsymbol{\eta}_t$ does not yield quantile bounds, these can, however, be added employing a now-standard method [Koolen and Van Erven, 2015]. Namely, instead of only including every expert with a private learning rate tuned to its prior complexity level (the typical $\ln K$ or $\ln(1/\pi_k)$ term), we include multiple copies of each expert, each with a learning rate tuned to a smaller complexity level. We then start from (7) with comparator distribution $\boldsymbol{p}$ concentrated on the $\varepsilon$-quantile of interest and carry out all future steps (from Proposition 2.2 on), ending up with the quantile regret bound $\langle \boldsymbol{p}, \boldsymbol{R}_t \rangle \leq \max_{k:p_k > 0} \sigma_k \sqrt{v_t (\ln C + D_{\boldsymbol{\eta}_0}(\boldsymbol{p}, \boldsymbol{u}))}$, where $C$ is the number of learning rates thus created. As these learning rates can be exponentially spaced in an interval of width $\ln K$, $C$ is of order $\ln \ln K$. Does this procedure answer Freund's question? For our notion of variance, $v_t$, which our results suggest is a rather useful notion, the answer is yes. However, to relate $\Delta v_t$ to $\mathrm{var}_{\boldsymbol{w}_t}(\boldsymbol{\ell}_t)$, we incur a multiplicative ratio $\eta_{t,\max}/\eta_{t,\min}$, which, for the quantile case, is of order $\sqrt{\ln K}$, turning the prior-in-the-square-root bound into a prior-outside-the-square-root bound. The latter was already achievable by not tuning $\boldsymbol{\eta}$ to the prior complexities at all. This problem does not arise in the same-scale uniform-prior case; there, $\Delta v_t$ is bounded by a small multiple of $\mathrm{var}_{\boldsymbol{w}_t}(\boldsymbol{\ell}_t)$ [De Rooij et al., 2014]. Note that this problem is present even when $K$ is fixed while $t$ grows, which is narrowly outside the scope of the impossibility results of Marinov and Zimmert [2021]. This discussion sheds light from another angle on why Freund's problem is hard; we present a desirable multiscale alternative.

**Luckiness, gap, and Massart's condition.** We now address the advantage of MUSCADA's refined second-order measure of time $v_t$ over the zeroth-order number of rounds $t$. Multiscale zeroth-order regret bounds (growing with $t$) can be guaranteed either by tuning MUSCADA crudely to a constant multiple of $t$ or by building an any-time improvement of the algorithm of Bubeck et al. [2019], also tuned to $t$. Both $t$-tuned and $v_t$-tuned algorithms have constant expected pseudoregret in stochastic lucky cases, but the constant can be widely different. Indeed, the constant for $t$-tuned algorithms scales with the inverse $1/d_{\min}$ of the gap $d_{\min} = \min_{k \neq k^*} \mathbf{E}[\ell_{t,k} - \ell_{t,k^*}]$, while the constant for $v_t$-tuned algorithms scales with the constant $c_{\mathrm{M}}$ from Massart's condition (see Definition 1.3). The difference stems from the fact that $c_{\mathrm{M}}$ is at most $1/d_{\min}$, but it can be arbitrarily smaller. This separation appears to be fundamental. In the single-scale uniform-prior case, the above $t$-tuned algorithms are closely related to Decreasing Hedge [Mourtada and Gaïffas, 2019], just as MUSCADA is related to AdaHedge (see Section 2.1). Mourtada and Gaïffas [2019] show that, in the single-scale case, even under Massart's condition with $c_{\mathrm{M}} = 1$, Decreasing Hedge and, consequently, Bubeck et al.'s algorithm with decreasing learning rates, has expected pseudoregret $\mathbf{E}[R_{t,k}^B] \gtrsim 1/d_{\min}$. If the smallest scale $\sigma_{\min} > 0$, by taking $d_{\min}$ small, this lower bound can be made arbitrarily worse than the guarantee of MUSCADA, $\mathbf{E}[R_{t,k^*}^{\mathrm{MUSCADA}}] \lesssim c_{\mathrm{M}} + 1$, from Theorem 3.1.

**Open problem.** Our ability to incorporate an arbitrary prior suggests that the results should extend to countably many experts. However, the current techniques do break down. When $\max_{k \in \mathbb{N}} \sigma_k < \infty$ MUSCADA with Tuning 1 (if $\inf_{k \in \mathbb{N}} \sigma_k > 0$) or Tuning 2 would still deliver the worst-case bound. Yet our luckiness result currently requires $\max_{k,l,t} \frac{\eta_{t,k}}{\eta_{t,l} \sigma_l^2} < \infty$. Even with a common scale $\sigma$, this is never the case due to the dependence of $\boldsymbol{\eta}_t$ on the prior $\boldsymbol{\pi}$, which is necessarily decreasing. Is luckiness actually possible, for example, in the online learning analog of the elegant challenge example presented by Talagrand [2014, Chapter 2]?

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
