# OpenReview forum: "Luckiness in Multiscale Online Learning"
_NeurIPS.cc/2022/Conference — NeurIPS 2022 Accept_

### Official Review · Reviewer_jvH3 · 2022-06-29

**Rating:** 7
**Confidence:** 3
**Soundness:** 4 excellent
**Presentation:** 4 excellent
**Contribution:** 3 good

**Summary:**

In multiscale online learning, the range of losses differs from an expert to another. People aims at designing algorithms with regret depending on the scale of the best experts instead of the maximum scaling. Several algorithms have already be proposed in the literature with worst-case analysis. However, none of them achieves a constant regret under Massart's condition. The authors introduce a new algorithm called Muscada which entails similar guarantees as previous algorithms in the worst case but also achieves constant regret in benign statistical case (Massart's condition). They also analyze an optimistic version of their algorithm and conduct numerical experiments.

**Questions:**

Would be interesting to compare numerically Muscada against FTRL with multiscale entropy (without your refinement).
Do you observe non-constant regret for other algorithms under Massart distribution ?

Typos:
l.232: incurrs -> incurs
l'281 an refinement -> a refinement

**Limitations:**

As discussed at the end of the paper, the main limitation is that it does not extend to countable may experts but it offers a good opportunity to future work.

**Strengths And Weaknesses:**

This work follows the line of research of FTRL with multiscale entropy regularization. The main contribution is to refine this method with corrections terms added to the losses and to give an analysis under Massart's condition. But it is not limited to this key result and explore several aspects of the proposed method with an analysis of several tunings, an optimistic version, a discussion with Freund's open problem...
The paper is well written. Theorems and proofs are clear and precise.
I checked only a tiny fraction of the proofs and arguments seems to add up.
The contribution is significant and the techniques developed are interesting. It is a nice contribution to the multiscale literature.

---

> ### Author Response · Authors · 2022-08-02
> **On FTRL with multiscale entropy**
>
> ### On FTRL with multiscale entropy:
>
>
> Other algorithms guarantee constant regret under a Massart distribution of the
> losses, but, in contrast to our algorithm, they do not simultaneously guarantee
> the multiscale min-max regret. For instance, as we argue in lines 81-93 of our
> paper, an algorithm of Chen et al.[1] would yield constant expected pseudoregret
> under a Massart distribution, but it does not guarantee the multiscale min-max
> regret. The currently available versions of FTRL with mulstiscale entropy of
> Bubeck et al.[2] does not achieve constant regret because it uses a fixed-horizon
> tuning, for which for which the lower bound under Massart losses is still √T (it
> would take √T rounds before the weights start to concentrate). From the analysis
> techniques of our work it would follow that a modified an anytime-tuned version
> of Bubeck et al.'s algorithm with ηₜ ∼ 1/√t also has constant regret under
> Massart noise.
>
> [1] Impossible Tuning Made Possible: A New Expert Algorithm and Its
> Applications, COLT 2021
>
> [2] Multi-scale Online Learning: Theory and Applications to Online Auctions and
> Pricing, JMLR 2019
>
> We thank for the spotting of typos.

---

### Official Review · Reviewer_eZu6 · 2022-07-07

**Rating:** 8
**Confidence:** 3
**Soundness:** 4 excellent
**Presentation:** 4 excellent
**Contribution:** 3 good

**Summary:**

The paper considers the setting of prediction with expert advice where the (known) loss ranges for each expert ($\sigma_k$ for expert $k$) may vary drastically (the “multi-scale” setting). They improve existing regret bounds that currently have the optimal dependence on $\sigma_k$ by replacing dependence on $t$ with a novel variance-like term $v_k$, which they show leads to constant regret in the multi-scale stochastic-with-a-gap setting. Until now, constant regret was only known in the single-scale case; the multi-scale case resulted in regret of size $\log(t)$. They also extend this to an optimistic variant, and apply this in the now-standard way to the game-matrix setting.

**Questions:**

1. If you pick an expert to compete against, the $\sigma_k$’s do not need to be known in advance. Can the authors confirm that there are *no* multi-scale results that similarly do not require $\sigma_k$ knowledge, and can they comment on whether relaxing the assumption of their prior knowledge seems possible?

2. Can the authors include in their simulations a comparison with multi-scale dependence on quantities that become small in in stochastic settings, such as the adaptive variants of PROD? This would be useful to see that they do not obtain multi-scale constant regret in the stochastic setting, and it is not just the fact that we don’t know how to prove they achieve this because of the limitations of Bernstein’s inequality.

3. The author’s arguments seem to rely rather heavily depend on limiting statements in $t$. It would be good if they can state some finite-time bounds (with additive errors that may be bounded as $t$ gets large) to allow for comparison with other finite time regret bounds for quantile regret, even if the main informal theorem statement is asymptotic in $t$ to highlight the salient terms.


**Limitations:**

I see no negative societal impacts of this theoretical work.

**Strengths And Weaknesses:**

Overall, the paper is very clearly written, provides a well-defined and useful contribution that I think will be valued in the literature, and does so using some new algorithmic and technical tools that can hopefully be reused. I read the proofs at a high level and did not find any issues. I have no major concerns, just a few questions for the authors that may help place this work in the literature slightly more clearly and provide a picture of what the follow-up works might be.

---

> ### Author Response · Authors · 2022-08-02
> **On limiting-t statements, expert scale knowledge and PROD-style algorithms**
>
> ### On limiting statements on t:
>
> In order to introduce our results as succinctly and clearly as possible, we used
> limiting statements in t for the introduction. However, Proposition 2.3 shows the
> finite-time regret bounds for MUSCADA with explicit additive errors. Right next
> to it we show how they imply the statements made in the introduction. In a
> nutshell, all time-dependent terms in the regret of a specific expert are either
> premultiplied by its own scale or the minimum scale among all the experts; there
> are no problematic terms.
>
>
> ### On knowledge about the scales of the experts:
>
>
> In the tunings of the algorithm that we propose (see Figure 2), the scales
> $\sigma_k$ of all the experts are assumed to be known. However, as we will
> comment, in preliminary versions of the tunings, we considered alternatives that did not require this knowledge but incurred in a cost that we
> deemed unattractive. Comparison to existing
> multiscale algorithms does not seem possible: none of them guarantees
> simultaneously the min-max worst-case regret and constant regret in stochastic
> lucky cases. Despite this, we will comment on existing algorithms that give
> weaker guarantees.
>
> In preliminary versions of MUSCADA we considered tunings that only needed the
> knowledge of largest scale $\sigma_{max}$; however, this may come at a cost. These
> trade-offs are inherent to the nature of the algorithm design. Here we mention
> two possible modifications.
>
>   - A modifications of Tuning 1 in Figure 2 (requiring $\sigma_{min} > 0$) that uses
>     $u_k$ proportional to pi_k ($\sigma_{min} / \sigma_{max}$) and $\gamma_k$
>     proportional to $\ln(1 / u_k)$ would not need knowledge of the scales. The
>     cost comes in the leading constant on the dominant term of the regret bound,
>     which would become prior-dependent.
>
>   - A modification of Tuning 2 in Figure 2 (that might have $\sigma_{min} = 0$) that
>     uses alpha_k to be a constant over experts would result in a regret bound of
>     the order $O(\sigma_k \sqrt{v_t \ln v_t} + \sigma_{max} \sqrt{v_t /
>     \ln(v_t)})$ as $v_t\to\infty$ (recall that if $v_t$ remains bounded, so does the regret). This regret bound is still of the sought
>     asymptotic order $O(\sigma_k \sqrt{v_t \ln v_t})$ but it might take a large
>     number of rounds before this term becomes dominant, specially when the scale
>     of the expert under consideration is very small relative to $\sigma_{max}$.
>
> Among the algorithms that have weaker guarantees, the algorithm of Chen et al.
> [1] (that only guarantees the min-max worst-case regret) only assumes the
> knowledge of $\sigma_{max}$. Their analysis also shows a lower-order
> time-dependent term premultiplied by $\sigma_{max}$. On the other hand, the
> algorithm of Bubeck et al. [2] assumes the knowledge of all scales.
>
>
> [1] Impossible Tuning Made Possible: A New Expert Algorithm and Its
> Applications, COLT 2021
>
> [2] Multi-scale Online Learning: Theory and Applications to Online Auctions and
> Pricing, JMLR 2019
>
>
> ### On the relation to PROD-style algorithms:
>
>
> Using an adaptive PROD-style like ML-Prod could guarantee constant regret in
> stochastic lucky cases, but it would fail to guarantee the min-max multiscale
> regret, which is at the center of our attention. The reason for this is that
> the second order correction that is implicit in the PROD-style algorithms that
> yield constant regret in stochastic lucky cases does not respect the scales of
> each expert.

---

> > ### Comment · Reviewer_eZu6 · 2022-08-05
> > **One more minor clarification**
> >
> > Thanks to the authors for their responses.
> >
> > Apologies, my review was not precise enough: I was referring to the limiting arguments in (the proof of) Theorem 3.2. I guess that the authors can state finite-time stochastic bounds extracted from intermediate steps in the proof, just that these are not very readable; but it would be great if this can be confirmed.
> >
> > I appreciate the clarifying discussion about knowing the scale in advance, I think this would be helpful to briefly comment on in the camera-ready version. Similarly for why Prod algorithms are not minimax optimal. I have no further questions for these areas.

---

> > > ### Author Response · Authors · 2022-08-08
> > > **Finite-time stochastic bounds are possible**
> > >
> > > Thanks to the reviewer for the clarification.
> > >
> > > The ingredients needed to prove finite-time stochastic bounds are present in the proof of Theorem 3.2; a modification with known techniques would suffice. Our proof uses the existence of a time $t^\star$ when the weights are sufficiently concentrated in expectation on the best expert.  A modification of our proof could give high-probability bounds for an analogous random time $T^\star$  that marks when the weights are sufficiently concentrated on the best expert. This would, in a similar fashion, imply high-probability regret bounds.
> > >
> > > We highlight that the technical difficulties that are faced, and the reason for the limiting statements as $d_{min}\to 0$, come from the main features of the algorithm: the dependence of the weights on the expert-dependent corrections $\mu_k$ and the implicitly defined normalization constant $a_t$.

---

> > > > ### Comment · Reviewer_eZu6 · 2022-08-08
> > > > **No further questions**
> > > >
> > > > Great, thanks for elaborating! I have no more concerns or questions.

---

### Official Review · Reviewer_R47f · 2022-07-11

**Rating:** 6
**Confidence:** 2
**Soundness:** 3 good
**Presentation:** 2 fair
**Contribution:** 3 good

**Summary:**

The paper extends online prediction with expert advice to a multiscale setting, where the experts' losses have different ranges. The proposed MUSCADA algorithm simultaneously guarantees the minimax regret that grows with the scale of the best expert and has a constant expected pseudo regret in the stochastic lucky case.  MUSCADA refined the Follow The Regularized Leader (FTRL) with a multiscale entropy regularizer, which has a second-order variance-like adaptation. Regret bounds for MUSCADA under two tuning schemes are further provided. Moreover, the authors build an optimistic version of MUSCADA and apply it to solve multiscale zero-sum games, then further experimentally demonstrate its superiority compared to the single-scale algorithms. Finally, the results are discussed with Freund's 2016 open problem.

**Questions:**

I suggest the authors modify the paper structure for better presentation.

**Limitations:**

Yes, the limitations are discussed in the open problem section.

**Strengths And Weaknesses:**

### Strengths

The proposed algorithm, tuning schemes, and theoretical results seem interesting and novel. The algorithm does not need knowledge of the game's length in advance and does not need to store any doubling trick. Stochastic luckiness is studied with quantified data easiness. And the constant regret for easy data is witnessed in the experiments on synthetic data.
However, I am not an expert in the related domain, and I did not carefully check the proof, so I cannot precisely assess the quality and significance of this work.

### Weaknesses

1. The structure of the paper causes some difficulties in understanding

Section 1.1 presents the results before the introduction of MUSCADA with some notations and concepts unknown.

2. The experiment setting is unclear, and some important descriptions are put in Appendix.

---

> ### Author Response · Authors · 2022-08-02
> **On the presentation of the article**
>
> ### On the presentation of the article:
>
> In Section 2 we described and motivated all the choices made in the design of
> MUSCADA, our main algorithm. There, we place MUSCADA in the context of other
> existing algorithms.  As pointed out already by Reviewer iHr2, we also included
> in Section 2.1 an instance of MUSCADA where all the quantities involved can be computed
> explicitly for the sake of clarity. We believe that giving more details about
> the inner workings of the main algorithm too soon will only result in a more
> confusing presentation.
>
> All the details of the experiments are laid out in detail in Appendix 1.

---

### Official Review · Reviewer_iHr2 · 2022-07-12

**Rating:** 5
**Confidence:** 3
**Soundness:** 3 good
**Presentation:** 3 good
**Contribution:** 2 fair

**Summary:**

This paper studies algorithms for the online prediction with experts problem in the multi-scale setup where the losses of the different experts have varying (and unknown) scale parameters. Within this setup, the work looks for an algorithm which is adaptive to the simpler stochastic (or lucky) case. The solution is an algorithm, called MUSCADA, and its regret analysis in both the worst-case and stochastic setup. The paper also presents an optimistic version of this and applies the resulting algorithm to zero-sum multiscale games.

**Questions:**

- Is there a lower bound for the algorithm proposed by Bubeck et al. 2019 under the Massart's easiness condition which shows that it cannot obtain the constant regret bound?
- What are some example applications where multiscale zero-sum games and the techniques proposed in this paper would be useful?

**Limitations:**

The paper is a theoretical work and does not have potential social negative impact. It clearly mentions the assumptions under which the results hold and also provides several directions for future work.


**Strengths And Weaknesses:**

The paper proposes a simple algorithm based on Follow the Regularized Leader approach and analyses its regret bound. The overall algorithm and its intuition is well explained as are the problem setup. The paper even instantiates the results for a simple example, that of single scale uniform prior, to provide some intuition about the algorithm.

In terms of techniques, the paper does not bring out the challenge and novelty which prohibit existing techniques from working for this particular setup. For instance, would a simple application of the Chen et al 2021 to Bubeck et al.'s 2019 algorithm work or is something more nuanced required here?

---

> ### Author Response · Authors · 2022-07-29
> **On Chen, Bubeck, Massart and applications.**
>
> On the relation to Chen et al 2021:
>
> We discuss the relation to Chen et al. in lines 81-93 of our paper. Chen et al. develops two versions of their algorithm, one with multi-scale worst-case regret, and one with stochastic luckiness for same-scale problems. Using their technique to achieve stochastic luckiness unavoidably loses the scale adaptivity, while our innovation achieves it. More precisely, Chen et al. and our papers are both based on second-order corrections to the loss, but these are of fundamentally different type. Chen et al.'s corrections are of the ML-Prod [1] type: they measure the difference between the expert loss and the learner's loss: as such they take values on the largest possible range. This is the reason Chen et al. can only show stochastic luckiness in the same-scale case. Our main innovation is to craft a different correction, which is of the AdaHedge [2] type; it is the correct multi-scale analogue of the variance of the expert loss under the learner's weights.
>
> [1] A Second-order Bound with Excess Losses, COLT 2014
> [2] Follow the leader if you can, Hedge if you must, JMLR 2014
>
>
> On the Massart condition:
> Bubeck et al. use fixed-horizon tuning η ∼ 1/√T, for which the lower bound under Massart losses is still √T (it would take √T rounds before the weights start to concentrate). The techniques we develop to prove Theorem 3.2 now make it possible to show that an anytime-tuned version of Bubeck et al.'s algorithm with ηₜ ∼ 1/√t also has constant regret under Massart noise. This was not previously known.
>
> On applications:
> One can easily imagine games with risky and safe options, where the loss range of the risky options is much larger than that of the safe options. Our algorithm then computes saddle points quicker if they turn out to use only safe options.

---

> > ### Comment · Reviewer_iHr2 · 2022-08-08
> > **response**
> >
> > Thanks a lot for your rebuttal. The authors comment that the algorithm by Bubeck et al also achieves constant regret under Massart noise but needs their analysis -- what is the advantage of the proposed algorithm in the paper in this case? Why not just analyze the existing algorithm by Bubeck et al.? Also, I am really not convinced about the response to the application side of things. It would be useful to provide more details on the particular application the authors have in mind and how their proposed algorithm would solve it better than existing results.

---

### Meta-Review · Area_Chair_Yybp · 2022-08-23

**Recommendation:** Accept
**Confidence:** Certain

**Metareview:**

The reviewers were generally happy with this paper. There were some comments about better experiments and one comment about properly comparing this work to Bubeck et al (e.g. difference in rates and comparative advantages). I encourage the authors to better explain this related work and also incorporate some of the clarifying discussions with reviewers in the final version of the paper.

**Award:**

No

---

### Decision · Program_Chairs · 2022-09-14

Accept